# 'I'm not a smoker…yet': a qualitative study on perceptions of tobacco control in Chinese high schools

Xiang Zhao,[1] Ross McD Young,[2] Katherine M White[1]

[1]School of Psychology and Counselling, Institute of Health and Biomedical Innovation, Queensland University of Technology, Brisbane, Queensland, Australia
[2]Faculty of Health, Institute of Health and Biomedical Innovation, Queensland University of Technology, Brisbane, Queensland, Australia

**Correspondence to**
Xiang Zhao;
xiang.zhao@hdr.qut.edu.au

## ABSTRACT

**Objective** Chinese adolescents' perceptions about tobacco control at schools are rarely researched. We explored how current antismoking strategies work in middle school environments, as well as the attitudes towards these strategies held by students and teaching staff members.

**Methods** Four focus groups (24 eleventh graders; $M_{age}$=16 years) and five indepth interviews (teaching staff members with tobacco control experience in schools) were conducted in Kunming, Yunnan Province, China. We used thematic analysis combining inductive and deductive processes, along with field observations and research group discussions, for data analysis.

**Results** With educational approaches and practical strategies, antismoking education reported in the middle schools had limited effectiveness. Although smoking is banned in schools, students can circumvent schools' controls easily. Notably salient is the pessimistic attitude towards school-based antismoking strategies at school. Detrimental influences within (teachers' smoking) and beyond schools (high societal smoking prevalence) largely challenged the efforts to manage students' smoking.

**Conclusions** Current antismoking approaches in schools fail to curb smoking among Chinese high school students. Their effectiveness is undermined by both within-campus and off-campus influences. Students' perceptions of smoking should be valued as their knowledge of smoking is actively constructed. Future antismoking education at school should incorporate interactive sessions rather than merely didactic approaches about the harms of smoking. Although stricter rules for teachers' smoking are needed, complementary strategies such as population-level interventions and policy measures in wider society will assist in efforts within schools.

## INTRODUCTION

### Background

China is the world's largest tobacco consumer. It currently has 316 million smokers (current smoking rates: male 52.1%, female 2.7%), and its adolescent smoking rates have increased in the past three decades.[1 2] This trend is alarming because smoking in adolescence is a strong predictor of heavy smoking in adulthood.[3 4] According to a meta-analysis, current smoking rates of male and female

## Strengths and limitations of this study

► This is a novel qualitative study investigating the tobacco control approaches available in Chinese middle schools, which are insufficiently researched despite their importance in terms of the current and future health of students.

► Opinion among both students and staff featuring in the same study is a strength of this work.

► The results of the study should be considered acknowledging its limitations, including the sample size and the single geographical location for the research site.

adolescents were estimated to be 17.4% and 3.26%, respectively.[1] Although China ratified the WHO Framework Convention on Tobacco Control as early as 2005, due to the interference by tobacco companies and lack of cross-ministerial cooperation for implementing the treaty, tobacco control in China has had little success.[5 6] Given a drastic increase of smoking rates is seen from mid-adolescence and late adolescence to the early 20s in several national surveys in China,[7] tobacco control is needed targeting high school-age adolescents.

Schools appear to be an ideal environment for tobacco control due to the relatively low cost and ease of implementation.[8 9] With 66.9 million secondary school students currently in China,[10] school-based programmes have the potential to prevent smoking efficiently. Nevertheless, despite government enforcement of antismoking policies in schools, teenage smoking has not been curbed[1] and effective interventions are still scarce.[11–13] Previous school-based smoking interventions were mostly undertaken in the West, especially in North America; generalising this experience to China is questionable.[9] Moreover, without students' evaluations using focus groups or surveys, the mechanisms of the better outcomes shown in some types of smoking

interventions remain unknown.[9] Given the novelty and necessity of this research topic in China, indepth investigations are warranted to assist in understanding what aspects of school-based interventions may be the most effective.

Chinese adolescents' perceptions of smoking interventions have not been examined in studies to date. Citizens in leading tobacco-producing areas (eg, Yunnan Province) have a high smoking prevalence as tobacco consumption is deemed to help the local economy,[14 15] and social practices such as tobacco gifting and offers constitute a stumbling block for smoking cessation.[16 17] As for high school students, since they are facing the competitive national Entrance Examination at the end of their final year at school,[18] tobacco control is important as smoking can be a means of coping with academic stress among Chinese teenagers.[15 17 19] Specific to school environments, tobacco retail sales, which are officially banned within 100 m of schools, are not fully enforced,[20] including in Yunnan Province.[14] Establishing the perceptions of young people in such high-risk environments on how to best combat smoking is crucial for better smoking intervention designs in the future.

Our research focused on Chinese schools, which contain some status-related power differentials (ie, teacher–student relationships) relative to antismoking management. Similar to the West,[21] the student–teacher relationship in Chinese schools appears to be an 'us-versus-them' structure, which is underpinned by traditional Confucian culture[22] and the contemporary Marxism-based[23] national school moral education system.[24] Distinguishable from the West, however, teachers in China are regarded as a model of both knowledge and morality.[22] Partly due to the respect for teachers, compared with disciplinary approaches used in other countries, Chinese teachers tend to be lenient and supportive even when students misbehave.[25] Moreover, all secondary schools in China follow the national outline of a moral education system,[24] an omnibus educational programme including ideological, health and other aspects; according to which, collectivism (eg, to respect others, to contribute one's strength for the community, to handle the relations between individual and collective interests) is highlighted as an important aim to be achieved during middle school education. The Outline also specifies that form teachers (also known as 'class teachers') and the head of the Teaching and Discipline Office play decisive roles in cultivating students' ideological and moral characters, as well as healthy habits (eg, not to smoke). In this sense, Chinese schools, although with huge student numbers and regional differences, can be regarded as organisations guided under a unitary moral education system. Given the dramatically different power belonging to students (objects of cultivation) and teachers (subjects of cultivation), analysing perceptions of both populations can serve to deepen the understanding of tobacco control at schools.

Given the absence of strong findings for antismoking interventions among Chinese high school students, the aim of this paper is to investigate the perceptions about school-based tobacco management among students and teaching staff members in high schools, in the hope of informing future research and practice. Our objective was to gain an understanding from students and teaching staff members in terms of the following research questions: (1) How do antismoking strategies work at school? (2) What are the attitudes towards antismoking approaches at school? (3) What approaches to smoking management are regarded as ideal at school? We designed the above research questions based on our eclectic philosophical position: students perform their smoking-related actions as rule instructors at school; such actions are also knowledge-constructing activities. In other words, the antismoking perceptions that students possess are regarded as both the knowledge students receive from school policies, as well as the knowledge they create through discourse.[26 27] For this reason, tobacco management at school is a dynamic process where students are passively following the rules and acting out their perceptions of the rules. Unlike positivistic research, our study did not aim to test predetermined hypotheses or create generalisations, but to holistically understand the intricacies of tobacco control in Chinese school contexts; thus, qualitative approaches were adopted as they are suitable for initial explorations of smoking-related perceptions, particularly among young people.[28–30]

## METHODS

### Sample

The research location was Kunming, a leading tobacco-producing region in China (for more details, see Field observations section). Given the research question is multilayered, we used mixed methods with various samples to identify the factors that serve as facilitators/barriers for antismoking education at school; this approach potentially avoided bias in homogeneous sampling.[31] Our qualitative data were generated from (1) four focus groups with six students in each group (three male and three female); and (2) semistructured interviews with five teaching staff members (three school teachers and two dormitory managers). Twenty-four students were recruited from four classes in two high schools irrespective of their smoking experience. Using a convenience sampling method, all student participants had previously completed three-wave surveys about smoking; two focus group members had received a four-session smoking intervention delivered by the research team (the team comprised one male PhD student and two professors, and all members had qualitative research experience on this research topic; a brief evaluation of the intervention was conducted among participants who had intervention experience as part of this research, and is reported elsewhere). Student participants were recruited by the researcher at the end of the third wave of the survey. There was no inclusion criteria (eg, smoking experience) for student volunteers, and all participants who previously completed questionnaires

were given the opportunity to partake in the interviews. A purposive sampling method was used for selecting relevant teaching staff members. Two form teachers whose classes participated in the intervention were invited to participate, and the principals provided names of other staff members with relevant experience in the context of tobacco control in the school. The other three teaching staff members included one head of the Teaching and Discipline Office and two senior dormitory managers. Following the Outline,[24] all interviewees were involved in smoking monitoring and control among students, as well as discussion with students who were caught smoking at school. To avoid identifying the informants, we only use 'form teacher' and 'staff member' at the end of the quotes.

## Data collection

The study used several approaches, including focus groups, interviews, field observations and research team discussions, to better comprehend the social settings surrounding school-based smoking.[32] Focus groups were chosen because this approach encourages all participants to express their opinions.[33] As opposed to individual interviews, focus groups tend to generate more sensitive and personal disclosures for health-related topics[34]; practically, as smoking is forbidden in schools, school-based individual interviews on smoking topics might resemble interrogation (especially for students who smoke), which may further discourage free discussion.[15] Triangulating data from different sources is especially important for our research as smoking at school is banned and participants might be reluctant to state their opinions due to this school policy. To manage possible social desirability, the following strategies were undertaken to encourage free expression of ideas: the interviewer emphasised the confidential nature of the research and requested that participants not share information (eg, smoking experience) they heard from other interviewees. For teaching staff members, we provided each of them a copy of their interview recording so that they could inform the researcher not to report some quotes or entirely withdraw their participation (although no participant contacted us). Due to the distinctions (eg, power, knowledge, age) between teaching staff members and students,[21] analyses of the contrasts enable the identification of central themes across heterogeneous samples.[31] Given the nature of our research questions, three or four focus groups were deemed as sufficient to achieve data saturation, as suggested by Krueger[35]; clear patterns appeared after the third interview among teaching staff members. It also should be noted that phenomena, instead of statistical inference, were the focus of this research. Therefore, using predetermined sample sizes to draw statistical inferences is not the aim of qualitative research.[31]

Form teachers provided a quiet environment for focus groups, typically a classroom. Before data collection, participants were informed about the confidentiality of their data. Other people were not present when the interview/focus group was conducted. To compensate participants' time, we gave a notebook (approximately US$5) to each student and a cash payment (approximately US$15) to each teaching staff member. Three teaching staff members completed the interview in the Kunming dialect as they felt more at ease; all others were in Chinese Mandarin. Dialogue was audio-recorded and translated verbatim into English. The first author (who grew up in a Kunming dialect-speaking area, received education in Chinese Mandarin and is currently undertaking a PhD in English) completed and checked the translation; epistemologically, this researcher/translator dual role could strengthen the rigour of research as the study was conducted with, from and inside the language by a community member.[36] Several group discussions with other authors (native English speakers) were used to further understand similarities and differences, linguistically and culturally. Each interview/focus group lasted for about 1 hour. All participants completed the interview/focus group. Two teaching staff members chose to receive a copy of the audio recording of their own interviews, but no further comments/corrections were returned to the research team. The above work was conducted by the first author. Generally, participants freely expressed their ideas in both interviews and focus groups; answers seemed genuine and natural. Although participants were not formally asked about their smoking status, both students and staff members frankly shared smoking-related experiences of their own or of friends and family members during the interviews.

To address the research questions, the research team developed the following general questions to elicit factors that may have facilitated or hampered achievement of the desired outcomes of school-based smoking programmes: (1) What antismoking approaches are available at school? (2) How do they work? (3) How do you evaluate these approaches? (4) How will you improve the tobacco management at school? These questions were used consistently in all focus groups/interviews. Questions in the guidelines only served to stimulate open discussions; follow-up discussions were further probed based on participants' responses. Before the data collection, several pilot interviews with older teenagers at the research site were conducted. No demographic or smoking-related information was collected from interviewees. The principals of the participating schools reviewed the research plan, including the ethical components of the research, and provided consent to undertake the study. Form teachers also gave their consent for students to partake in the study. All participants signed consent forms.

## Field observations

The present study was conducted in Kunming, the capital city of Yunnan Province and the key tobacco-producing region in China. The tobacco industry constitutes a substantial part of the local economy. During the field trips, public smoking was prevalent indoors and outdoors. Few places have strong smoking prohibitions,

except for schools and petrol stations. Middle school students smoking on campus is not commonly observed as it usually occurs in hidden places (eg, toilets). I (the first author) visited the male toilets in both schools and saw cigarette butts on the floor. During break times, I saw some male students gathering together and smoking. They appeared astonished at first when they saw me as they thought I was a teacher from the school and might punish them. Teachers' smoking was witnessed in both schools. One or two posters with no smoking signs were seen on the campuses. Interestingly, during the field trips, local television programmes reported several events where Kunming citizens who asked smokers to stop smoking in lifts or bus cabins were physically attacked by other smokers.

## Data analysis

Data were analysed thematically.[37] Three researchers independently read the transcripts. The first author coded initial categories/themes from both focus groups and interviews. Themes across the data set were collectively discussed and refined over several meetings, and invariant themes across data were synthesised as final themes.[38] Then, the first author reviewed the representativeness of themes and selected quotes. The analysis was finalised after several group discussions and revision. Three themes were identified across different samples as described in the following section. The present paper followed the guideline of the Consolidated criteria for Reporting Qualitative research.[39]

## RESULTS

### Tobacco control systems at school

The first theme is a descriptive summary of tobacco control system identified at two schools. Although the theme is mainly based on the narratives of teaching staff members, cross checking with data from student samples was also conducted. To retain thematic cohesion, the probing of these school policies is elaborated in the second and third themes.

All students are educated that smoking is harmful to their health. Schools provide this education using several methods including blackboard displays, theme class meetings and speeches under the national flag. The content is mainly about the negative outcomes of tobacco smoking. Visual materials showing the toxicity of nicotine were regarded as influential for students.

> I once asked form teachers to play a video during their theme class-meetings; the video is an experiment which shows the harm of one cigarette's nicotine to a mouse. Form teachers were asked to lead related discussions with students after watching the video. (Staff member)

If students are found smoking on the campus, form teachers will summon the parent(s) to school and tell them the situation and emphasise the antismoking

policies at school. Additionally, form teachers will conduct 'ideological work' with the student one-on-one. The ideological work is an all-purpose method to deal with various problematic students in Chinese schools[24]; it aims to let the student know a certain behaviour is wrong and, thus, to correct it. Rather than targeting a specific behaviour (eg, not to smoke), the ideological work compels students to obey the rules (ie, smoking is banned, therefore one should not break the rules by smoking).

> [If] a student has a problem, then the form teachers should talk to his or her parent(s) in order to know their family background, the student's family behaviour, and the parents' attitudes. (Staff member)

> …ideological education is more important…You have to let them know it is a wrong thing, as well as to remember it is wrong. The most important thing is to make students aware of the facts and reasons…I firstly talk to them and then let them write a guarantee showing his/her understandings of the issue—why it is a wrong thing, how to rectify it. (Form teacher)

Teaching staff members lacked effective measures to deal with students who frequently smoked at school. The Teaching and Discipline Office plays an important role in dealing with these difficult cases. Depending on the seriousness of the case, the office would issue a demerit (from minor to major), send the student back home to give up smoking or expel the student. However, schools rarely expel students due to their smoking even if it is serious. As some staff members reported, this inability to expel students makes tobacco control difficult at school. Similarly, if a demerit is issued, this record may be written into the student's archive (a Chinese system that employers can scrutinise); practically, the teaching staff members we interviewed in this study reported that demerits will often be retracted before the student graduates.

> We cannot expel students or persuade them to quit school because they smoked. Especially during the compulsory education stage [from 1st to 9th grade], no student can be expelled; students in that stage have rights to receive education—such rights are protected in Education Law. Although high school students are not in the compulsory education stage, expelling them if they smoked will cause heaps of troubles. (Staff member)

> I have not heard of any student's misbehaviour being written in their Archives. (Form teacher)

Apart from the above measures, several auxiliary approaches are used. Teaching staff members often patrol the dormitory passages and monitor the male toilets. When students return to school, security guards routinely check students' bags to ensure that forbidden objects including tobacco are not brought onto campus. Interclass competitions were also used, with smoking incidents in a class resulting in deductions of points for the class.

Students have to restrict their [smoking] desire till they leave the campus. However, in the morning, I at times pick up smoky smells in some rooms. In such cases, I will deduct the scores for that room and address students in the following noon break time. (Staff member)

## Challenges and mistrust of antismoking strategies

The management approaches described by staff were confirmed by students, but several issues seemed to prevent tobacco control from functioning properly. First, carefully monitoring a large number of students is impossible. Patrolling and bag control appeared to be ineffective as students could bypass those measures. Surprisingly, some students even reported that parcels were used to deliver tobacco to their dormitory; since a parcel is a personal property, schools could not check the contents. Teaching staff members also acknowledged that buildings are too large to be closely monitored.

You can never stop this. You think we are not likely to smoke at 3am or 4am, but we do it [in the dorm]. We observe the pattern—we smoke when we feel they [dorm staff] will not appear. (Male student)

I know some students separated a pack of cigarettes into single ones and hid them in different places such as pencil cases. (Male student)

In the teaching building, the space is big, it is impossible to monitor smoking in every corner. (Staff member)

Second, an inaccurate understanding of smoking was prevalent throughout the discussions. For students, the harm of tobacco was underestimated. Some students thought smoking was normal during adolescence, assisted coping with stress and helped the economy, and occasionally reported that smoking has benefits for one's health. In contrast, teaching staff members all acknowledged that smoking is harmful to one's health. Nevertheless, they agreed with most of the functions of tobacco use reported by students. Additionally, teaching staff members often regarded smoking as purely a psychological dependence. Even one of the form teachers who teaches biology did not think that tobacco addiction might require medical treatment.

I do not think smoking can have an impact on the country. Smoking adds tax income for the country. Even if it is at war time, soldiers who smoke will not be a problem. In recent decades, almost every soldier smokes; Chinese soldiers now are mostly smokers. Their combat ability and health is not weaker. So, I think smoking will not influence the country. (Male student)

My mother works in a hospital and my grandpa was an in-patient there. I found [in that hospital], when a patient is badly ill, the doctor would comfort the patient with some toxic material. So, smoking should

be like that; it helps people to deal with their pain…I think smoking is both good and bad. It helps people to cope with stress. Smoking moderately will not harm people. (Female student)

Smoking can reduce stress, but we still need to educate students. They have other ways to reduce stress. For example, sports, chats, basketball matches, art festivals. (Staff member)

How can we categorise it [smoking] as a serious problem as the tobacco industry is still running and cigarette trading is legal in the country? You know, our nation is still making the 'Great Zhonghua' ['Zhonghua' is a pun: it refers to a famous Chinese cigarette brand as well as the literal meaning, 'China']. We get big money from Zhonghua cigarettes and foreigners are fond of it. (Staff member)

Third, the effectiveness of antismoking education was doubted by both students and teaching staff. Instead of health promotion, safety was the ultimate reason behind tobacco control at school as smoking causes fires, according to teaching staff members:

Kids put the lit cigarettes in the dorm and they might cause a fire in the room. Safety matters. Some students craving a cigarette might light a cigarette and burn the beddings and himself/herself. So, smoking cigarettes is not allowed. (Staff member)

Both students and teaching staff held pessimistic attitudes towards smoking interventions. Health education, along with ideological education, was regarded as unlikely to be effective. Being an appropriate age and under heavy academic pressure were reported as justifications for smoking, especially among boys.

It is like a norm that most boys who are 16 or 17 years smoke. So, with intervention programmes, it is hard to control tobacco use. (Female student)

Oh, my! You are too naïve. They [smoking interventions] definitely cannot control smoking…students are facing huge academic pressure, especially 12th graders. You ask them not to smoke?—no way! (Staff member)

Speaking of ideological work, its effect is like the outcome of health education—not much effect. The form teacher did their work, I thought the content of the sermon was quite right, but after 2 hours or even just 2 min, I thought it actually was incorrect. (Male student)

The lampoon below from two male students in response to a girl's suggestion vividly shows students' attitudes towards antismoking education:

Female: Maybe designing and posting some powerful [antismoking] signs…

Male (1): They have no effect on people.

Male (2): Right. People won't read them!

Male (1): People will smoke even when they read them. Nobody can stop smokers. So, any sign is merely a sign.

Interviewer: Could any powerful signs or languages work at all?

Male (1): I think they are useless no matter how powerful they are.

Male (2): I will just think the sign is interesting and take a picture of it and post it on my WeChat Moments [a Chinese phone app, similar to Instagram and Facebook]. Maybe take the photo while I am smoking under the sign.

Although teaching staff members generally lacked confidence in proposing any practical approaches to manage student smoking, a few plausible strategies were reported by students such as an intensive smoke surveillance system, as well as separating smokers from non-smokers:

My junior middle school did have smoke detectors in every corner. Anyone who smoked will be caught at once. It is a very good solution. I also think that form teachers should separate smokers into different groups. If a dorm room has many smokers, those who do not smoke might become smokers soon. (Female student)

### Detrimental influences from wider society prompt smoking

During the field observation, shops selling cigarettes were easy to find around both participating schools. Students reported they were able to purchase cigarettes even as teenagers. Notably, in one school, students could buy cigarettes from a nearby supermarket with their smart cards (cards that parents deposit money in advance for students' daily expenses). Teaching staff members thought that restricting access to shops close to the school would be of little use as students could still get cigarettes from other shops slightly further away. Pocket-money control was referred to as a method to limit students' smoking, which was disregarded by another teaching staff member who stated that students have various ways of obtaining cigarettes such as asking for them from a friend.

Most shops sell cigarettes. Last time, when I bought something in a shop, I just glimpsed at the cigarettes. The shopper immediately asked me which type I wanted. (Male student)

They can still get cigarettes. You know, there are day students who can bring cigarettes to the campus… Even if you stop the supermarkets from selling cigarettes, students can still buy them from other shops beyond the school. So, the issue is uncontrollable. (Staff member)

The 'smoking world' beyond the campus was a big concern for both teachers and students. For teachers, they stated that their preventive work means little when influenced by students' family members. According to staff members, family was not the only source, but the whole society posed a risk in terms of smoking. When socialising with strangers, cigarette offers to alleviate embarrassment and bridge close relationships were commonly mentioned by both students and teachers, as exemplified in the quotes below. For this reason, male students reported that they might smoke in the future for better socialisation when they are adults, even though they did not smoke now as students. Although nearly impossible to stop, teaching staff members all thought that tobacco control at school was necessary. Concernedly, some approaches reported by staff members were likely to lead to future smoking among students.

We often feel that 5-day-controlling comes to naught due to their 2-day-home-staying. Their parents and their new friends can affect them. I feel that peer influence is larger than teachers' influence for these students. (Form teacher)

When you go to places where people sing karaoke, if you do not smoke there with them [old friends], they will think that you despise them, and you don't smoke like them because you are now in a good school. Then, they might end their friendship with you. In that case, you have to light your cigarette and smoke with them. (Female student)

When I say 'sorry, I'm not a smoker…yet', people will normally withdraw the cigarette. (Male student)

I will ask the student [who smoked] to go to my office…I will educate him as such: 'how dare you smoke? Smoking is firstly bad for your body. And it is not easy for your parents to earn money. When you enter society and you feel you are stressed, then you can smoke occasionally. But it is not allowed for you to smoke now'. (Staff member)

As a saying goes, 'tobacco and alcohol bring people together'. Strangers look friendlier when a cigarette is offered. (Staff member)

Even within the school campus, smoking influences existed. Students reported that they had seen teachers smoking in the campus, which was confirmed by all teaching staff interviewees. Some teachers even presented students with knowledge about the positive outcomes of smoking. Furthermore, students observed that people with authority smoked, such as soldiers smoking during military trainings. Interviews with teaching staff members agreed that there are teachers who smoke and that stringent antismoking rules should be stipulated at school so that staff are good role models for students.

I remember my form teacher in junior middle school told us that a successful man is abnormal if he does not smoke. (Male student)

The school should set up rules to deal with this matter [teachers' smoking]. Like what I said, teaching by setting yourself as an example is more important

than teaching by words, teachers cannot control students' tobacco use if they themselves are smokers… Students watch what teachers do. Sometimes, teachers asked students not to smoke with a lit cigarette in their mouth. It will only be less effective. (Staff member)

I think teachers' smoking in front of students is very bad. (Female student)

## DISCUSSION

This study serves as an indepth exploration among students and staff members about perceptions of health education-related smoking strategies in Chinese school settings. Combining both participants' perspectives as well as field observations, tobacco control at school is richly represented. The study highlighted the shared pessimistic attitudes towards smoking interventions, whose outcomes are undermined by social environmental factors beyond schools. In terms of tobacco management at schools, our findings shed light on the teacher–student structure by comparing discussions from both samples, providing implications for future antismoking strategies. To date, school-based antismoking programmes have failed to curb adolescent smoking in China, and findings from this study contribute valuable information for future tobacco control.

Two main strategies were identified in middle schools: health education and punishment-related contraventions of smoking-free policies; the latter approach was considered more effective. Other practical approaches such as patrolling are also reported. However, participants reported that both approaches failed to sufficiently curb students' smoking, especially for high school students who reported multiple strategies to circumvent the tobacco control efforts at school. These strategies to manage smoking at school are strongly influenced by moral education approaches. One example is the collective punishment (group demerit points). Driven by the aim of cultivating collectivism among students,[24] such an approach might not be suitable to shape students' self-disciplined health concepts. Consistent with previous educational findings, Chinese teachers in our research also tend to use lenient, inclusive approaches to deal with students' smoking behaviours at school; strategies included discussions and seeking support from parents.[23 25] These methods might work for other problematic behaviours, but, ironically, because many fathers are smokers in China, the above methods may be of little assistance to stop smoking. Obviously, both collective and individual approaches were ineffective; rather than using an omnibus method following the Outline,[24] it may be beneficial to design specific strategies for smoking behaviours targeting students who have difficulties with smoking cessation.

Two contexts appeared to be crucial to decipher the ineffectiveness of schools' tobacco control policies. First,

at a personal level, understandings of smoking and antismoking programmes included inaccuracies. Consistent with findings of other adolescent/youth samples, 'willpower' was believed to be more effective than antismoking programmes provided by schools,[40 41] and harm-related information was largely underestimated.[30] Some perceptions were likely to be underpinned by lay health beliefs such as tobacco's medical functions in traditional Chinese medicine.[42–45] Although antismoking knowledge is available at school, as it is driven by ideological/moral education-based approaches (eg, simply forbidding students to smoke), the health-related influences of smoking might be largely downplayed. Second, at a school-environment level, the one-sided smoke-free policy undermines the effectiveness of tobacco control: teaching staff members are privileged as they have elevated status with the special 'right'—although unsanctioned—to smoke on campus. The structural power between teachers and students at school is, therefore, likely to prompt students to challenge any health imperative from the school's authority (eg, looking for the loopholes in tobacco management).[21 46] This finding also helps to explain why previous studies identified the positive associations between teachers' smoking and student smoking.[8 47 48]

In this study, mechanisms that enabled tobacco control to be effective were only limited to the concern about safety. By contrast, social norms related to smoking were identified as a constraining mechanism for tobacco control at school. Participants reported that smoking outside of school campuses was ubiquitous and perceived as a useful social tool. As found previously, the smoking behaviour of parents and teachers can lead to adolescent smoking and prosmoking attitudes[49]; high acceptability and prevalence of smoking outside of schools also served as a barrier to smoking cessation.[50 51] Consistent with most smoking studies among Chinese secondary school students, coping with academic stress was mentioned by students and teaching staff members as a reason to smoke.[15 17 19] As reported by our participants, this stress is especially pronounced for high school students as they are facing the Entrance Examination.[18] Thus, although the current school-based tobacco control has room to improve, the social norms of smoking and huge academic pressure further diminish any health imperatives about smoking.

Findings from this study provide global implications for future research. Antismoking policy in Chinese schools is seemingly a well-structured system with education, monitoring and enforcement processes. However, consistent with evaluations of the effectiveness of tobacco management in the West,[8] the policy does not appear to be effective. Importantly, smoking intervention in China including school policy and parental modelling also failed to control middle school students' smoking initiation,[12] which again amplifies the fact that schools are not vacuums and smoking interventions should move beyond the individual level.[52] In light of the power structure between teachers and students in school contexts,

addressing teachers' smoking is important. However, given the high smoking prevalence in wider society perceived by both students and staff, policy intervention beyond schools is necessary to better support tobacco control at school. Findings from other Asian regions with high smoking rates among men showed that influences beyond school appear to be more impactful than those within schools.[53 54] As reported by students and informants, some well-reported functions of smoking such as an academic stress coping strategy might be also learnt by social osmosis (eg, media, social interactions) from wider society.[15 17 28] Since the effectiveness of school-based smoking interventions hinges on the social environment outside of schools, aggressive and comprehensive antismoking policies in wider society should be implemented.[6 20 47 55] In light of the high acceptance and prevalence of smoking in Chinese social milieu, developing and implementing programmes with community-based approaches and ecological approaches could be important complementary strategies for school-based interventions[56]. Given the interference from the tobacco industry (eg, leading advertisements),[5 6] multiministerial policy interventions should also be considered. Measures such as supply-side interventions and establishment of smoke-free areas could shape antismoking social norm and behaviours. Broad societal changes may be necessary before strategies can be successful in targeting individual cognitions.

Sampling is a potential limitation, and generalising the findings of the current study should consider contextual factors in a particular geographical area. Moreover, although we tried to limit social desirability influences, teaching staff members might have restricted their negative opinions about school policy due to their positions at school. Importantly, our study highlights that high school students obtain their knowledge about smoking in an agentic and active way rather than passively receiving education and rules. School tobacco management strategies may not result in successful outcomes if within-campus and off-campus influences remain.

**Acknowledgements** The authors appreciate all participants of this study for their time and thoughts, as well as the arrangements made by the school principals.

**Contributors** XZ was responsible for research design, data collection and analysis. RMY and KMW contributed to the initial methodology, and were involved in data analysis and group discussion. XZ wrote the first draft. RMY and KMW provided revisions to the draft.

**Funding** This research received no specific grant from any funding agency in the public, commercial or not-for-profit sectors.

**Competing interests** None declared.

**Patient consent** Obtained.

**Ethics approval** This research was approved by the QUT's University Human Research Ethics Committee (approval number: 1500001027).

**Provenance and peer review** Not commissioned; externally peer reviewed.

**Data sharing statement** No additional data available.

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
