## [Reviewer comments · BMJ Open]

ARTICLE DETAILS

TITLE (PROVISIONAL)	"I'm not a smoker...yet": A qualitative study on perceptions of tobacco control in Chinese high schools
AUTHORS	Zhao, Xiang; Young, Ross; White, Katherine

VERSION 1 – REVIEW

REVIEWER	Charles Saunders Florida State University College of Medicine USA
REVIEW RETURNED	22-Sep-2017

GENERAL COMMENTS	This original article examines adolescent tobacco smoking in China. Specifically, the authors investigate anti-smoking policies in schools and adolescent's perceptions of smoking and smoking interventions. The results of the study come from interviews with twenty-four students and 5 school teaching staff. Purposive sampling was used 'for selecting relevant staff members' and focus groups were used for student interviews. While the results of the article are of interest, it is difficult to determine how much validity can be placed on such as small, non-probabilistic sample, particularly when the target population is '66.9 million secondary students'. The authors acknowledge the sampling issue but this still does not negate the fact of the extraordinarily small sample size. Perhaps, research of this type is extremely difficult to accomplish in China yet the authors recommend 'longitudinal and other robust designs'. Frankly, the article reads more like example of what could be published in the popular press as it relies extensively on quoted conversations with the participants. Unfortunately, the article lacks rigor.
---

REVIEWER	Janet Hoek University of Otago, New Zealand
REVIEW RETURNED	26-Sep-2017

GENERAL COMMENTS	Thank you for the opportunity to read this interesting MS. I think you have examined an important topic and enabled readers from outside China to learn more about smoking prevalence and interventions
---

within this important economy.

I have some suggestions that I hope will be useful to you as you develop your MS.

First, I wondered if you could provide more detail in your abstract; as an international reader, it would be helpful to know the age rather than grade of your participants. It also was not clear what experience teaching staff had and your choice of location could have been clarified. The results section should separate findings from children and key informants and could you be more specific about terms such as “detrimental influences from the social environment” ? Your conclusions are also quite general – could you suggest some specific actions that could be taken?

The strengths and limitations are more like a summary of the project than an assessment of strengths and limitations; it might be worth reading recent BMJ Open papers to assess how other papers have addressed this section. Do be wary of generalising from qualitative findings; the first point seems to imply the findings “ have consequences” for secondary students in general whereas qualitative data are not usually used in that way.

For international readers, some more detail of smoking prevalence and smokefree initiatives in China would be helpful. You assume that targeting high school students will protect them from smoking initiation into young adulthood but evidence from other countries, including New Zealand, is that despite very low smoking prevalence among adolescents, smoking uptake occurs rapidly from age 18 and peaks among young adults. The longevity of school-based interventions needs stronger support (or qualification) and there are strong arguments for a more comprehensive approach that incorporates school based measures alongside population-level interventions. Could you develop the background to consider use of complementary strategies? I would also urge some caution about young people’s perceptions as the measures they support are not necessarily those they see as most effective. I also wondered if you could clarify your objectives, some of which seem to require a different design (e.g. (iii) seems to be better suited to a pre-post design).

Could you provide some justification for your use of focus groups with students – why not undertake in-depth interviews to elicit richer data, or use friendship dyads? Why select students who had previously participated in a longitudinal survey on smoking and some who had received a smokefree intervention? I can understand how the sample was convenient, but can you also explain how your approach provided a robust sample? How did you manage the social desirability error that you suggest might have impeded participants from commenting on their smoking status? How did you assess data saturation (four focus groups seems relatively few to achieve saturation)?

Could you also explain the ethics review process you used? How did you manage translations?

Can you be sure the people you interviewed had access to the information needed to answer your research questions (i.e., were teachers involved in evaluating smokefree schemes? Did they have input into the management of these?).

I was a little surprised that you combined data from students and

	teachers, given they have quite different knowledge, power and perspectives, it might be more profitable to analyse the samples separately? Some themes draw only on one group (e.g. the ideological approach) and the theme is very descriptive with little probing of the tension between possible actions (e.g. expulsion) and actual actions taken. There seems to be an emphasis on collective punishments (group demerit points) – could you comment on this approach compared to the individual interventions outlined earlier in the section? The theme on anti-smoking strategies not working seemed to cover several sub-themes, such as monitoring and enforcement; mis-understanding of smoking’s risks and nature, and collective vs individual interventions. Can you tease out these quite different ideas a little more and comment on their implications? Evidence that tobacco was easily available suggests supply-side interventions and smokefree area measures could be appropriate. Could you discuss policy measures and how these could parallel or support some of the individual approaches you outlined? Often environmental change will bring about changes in social norms because some behaviours will no longer be possible. Could you develop discussion of policy interventions that could better support the work teachers are doing in schools? Finally, as noted above, your discussion of limitations could be expanded to include topics other than sampling. I hope these comments are useful and I wish you well with future research.
--	--

REVIEWER	Kylie Morphet University of Queensland, Australia
REVIEW RETURNED	31-Oct-2017

GENERAL COMMENTS	This manuscript reports on a qualitative study of students and teachers at two Chinese high schools about school-based smoking policies. I found this manuscript very interesting and enjoyable to read. I really appreciated the field notes that gave a sense of context to the findings for those not familiar with the environment in relation to tobacco control in China. I thought the sample size and the methods were appropriate, though further detail could be provided about various methodological aspects (which I’ve specifically discussed below). The authors illustrate very well how difficult it is to implement school-based anti-smoking strategies when the surrounding culture is not conducive to this. They also provide an insight into the mistaken beliefs of young people about tobacco, who downplay the health harms of smoking, e.g., the student who says that “Smoking moderately will not harm people.” As the authors point out, it is very important to conduct research on young people in China and their attitudes towards smoking, given the high and increasing smoking rates in this country, and the huge number of people who will suffer smoking-related disease in the future. I think some aspects of the methods and results could be clearer and more focused, and have made suggestions below in this regard:  1. On Page 4, in the background section, the authors state that
---

prevalence of smoking in young people is increasing. It would be good to have some numbers here – what does the latest data say about the proportion of adolescents who smoke in China?

2. In the abstract, it would be helpful to add that Kunming is in Yunnan Province, as you talk about high smoking prevalence in Yunnan Province in the introduction, but many people won't know that Kunming is the capital of Yunnan (and you don't make this clear until Page 7).

3. Can you provide information about how the students were recruited? Did the teacher mention it to students and allow them to self-select, or did teachers/researcher select students based on certain characteristics?

4. Did you collect data on how many of the students and teachers were current smokers, or had tried cigarettes? This has obvious relevance for their attitudes, so it would be good to report, if you have this information.

5. Some sentences are unnecessarily wordy and could be amended for better clarity. For example the first sentence in the methods reads "We used mixed methods with various samples to identify the inter-woven factors embedded in multi-layered phenomena and potentially avoid bias in homogenous sample." This could be stated much more clearly. As another example, on Page 6, Line 50: "Based on our research objectives, the research team developed guidelines to elicit factors that may have facilitated or hampered achievement of the desired outcomes." I'm not clear here what "desired outcomes" refers to, or what the "guidelines" are in relation to.

6. Were the interview questions the same or different for teachers and students? I assume they would be different questions, as they are quite different populations. The "general questions" listed on Page 6, Line 55 seem to be appropriate to ask teachers, but not students. For example, it's not really relevant to ask students "How will you improve the tobacco management at school", as this is not their responsibility. Providing more details about the interview schedule for both teachers and students would be helpful.

7. Relatedly, when reporting results, it needs to be drawn out more clearly whether the themes being discussed are based on data from the teachers or from the students.

8. On Page 7, Line 10, does "late teenagers" mean "older teenagers."?

9. The concept of "ideological work" seems rather vague to me. From what you've described, it seems that the teachers have a range of strategies for dealing with smoking at school, including practical strategies such as patrolling dormitories and checking bags. There are also more educational strategies, which aim to teach children about the dangers of smoking, and also to educate them about the rules regarding smoking at school. I'm just not sure that concept of "ideological work" adequately covers the data that the authors have presented.

10. Based on the findings presented there, the theme "Anti-smoking strategies at school are unlikely to work" might be better reframed in terms of pessimism about school based anti-smoking strategies.

11. On Page 11, Line 44, the authors write that "The effectiveness of anti-smoking education was doubted by all participants. Instead of health promotion, safety was the ultimate reason behind tobacco control at school as smoking causes fires." It would be helpful here, and in the manuscript in general, to tease out a bit more which themes were raised by students and which were raised by teachers.

	These are very different population groups. So for example, instead of writing “All participants” you could write “Both students and teachers” to make it clear this was a theme for both groups. 12. How do cigarettes “alleviate embarrassment” (Page 13, Line 27). Can you elaborate on this? 13. Page 15, Line 3, missing word. Should read “As school-based anti-smoking programmes have failed...” 14. On Page 15, Line 12, it would be more accurate to write that “...participants reported that both approaches failed to sufficiently curb students’ smoking” as you are not actually measuring the schools’ anti-smoking impact on smoking, just people’s perception of it. 15. Page 15, Line 20, should read “the ineffectiveness of schools’ tobacco control programs.” Or policies. 16. On Page 15, Line 44, the authors write that “In this study, mechanisms that enabled tobacco control to be effective were only limited to the concern about safety.” There isn’t much data in the results about this. Could a quote or two be added to support this? 17. On Page 15-16 you talk about literature showing that school students smoke to cope with academic stress. This literature should also be mentioned in the introduction. 18. On Page 16, Line 20, the sentence “Rather than establishing strict tobacco-free regimes at school which are currently unavailable, studies with longitudinal and other robust designs should pilot the effectiveness of these policies, because most associated research was conducted in Western countries and the effectiveness of tobacco management remains inconclusive.” This sentence is convoluted and rather confusing. Can you reword for clarity? 19. On Page 16, Line 41 the authors write that “As previously found, the connection between students’ smoking and coping with academic stress may be socially constructed.” Can you elaborate more on what you mean by this? 20. On Page 17, Line 26 the word “manage” should be “management.”
--	--

VERSION 1 – AUTHOR RESPONSE

Reviewer 1

This original article examines adolescent tobacco smoking in China. Specifically, the authors investigate anti-smoking policies in schools and adolescent’s perceptions of smoking and smoking interventions.

The results of the study come from interviews with twenty-four students and 5 school teaching staff. Purposive sampling was used ‘for selecting relevant staff members’ and focus groups were used for student interviews.

R1.1 While the results of the article are of interest, it is difficult to determine how much validity can be placed on such as small, non-probabilistic sample, particularly when the target population is ’66.9

million secondary students'. The authors acknowledge the sampling issue but this still does not negate the fact of the extraordinarily small sample size. Perhaps, research of this type is extremely difficult to accomplish in China yet the authors recommend 'longitudinal and other robust designs'.

Frankly, the article reads more like example of what could be published in the popular press as it relies extensively on quoted conversations with the participants. Unfortunately, the article lacks rigor.

Response: We thank Reviewer 1 for the above comments. We have modified the wording of "have consequences for 66.9 million" as the original wording contain causal connotations. However, please also note that probability sampling techniques are not suitable to understand qualitative research. We also do not agree with the criticisms suggesting that qualitative studies are not rigorous. Several studies (cited) published in highly regarded journals such as Tobacco Control evidence that the tobacco research field values qualitative research. Some rationales of using qualitative research approaches have been emphasised in the 'background' section to clarify the important contribution of qualitative perspectives to this area.

Reviewer: 2

Thank you for the opportunity to read this interesting MS. I think you have examined an important topic and enabled readers from outside China to learn more about smoking prevalence and interventions within this important economy. I have some suggestions that I hope will be useful to you as you develop your MS.

R2.1. First, I wondered if you could provide more detail in your abstract; as an international reader, it would be helpful to know the age rather than grade of your participants. It also was not clear what experience teaching staff had and your choice of location could have been clarified. The results section should separate findings from children and key informants and could you be more specific about terms such as "detrimental influences from the social environment" ? Your conclusions are also quite general – could you suggest some specific actions that could be taken?

Response: Thank you for raising these issues. Age has now been included in the Abstract. Teaching staff members' relevant experience is added in 'Sample' section. Given the contrast between samples was the focus our research, we have remained the data under the same theme. However, more comparative analyses were added. "detrimental influences from the social environment" was reworded. More specific action points have now been added in the Conclusions section.

R2.2The strengths and limitations are more like a summary of the project than an assessment of strengths and limitations; it might be worth reading recent BMJ Open papers to assess how other papers have addressed this section. Do be wary of generalising from qualitative findings; the first point seems to imply the findings " have consequences" for secondary students in general whereas qualitative data are not usually used in that way.

Response: We thank the Reviewer for pointing out these issues. The strengths and limitations has now been updated. The wording of the first point was rewritten.

R2.3For international readers, some more detail of smoking prevalence and smokefree initiatives in China would be helpful. You assume that targeting high school students will protect them from smoking initiation into young adulthood but evidence from other countries, including New Zealand, is that despite very low smoking prevalence among adolescents, smoking uptake occurs rapidly from age 18 and peaks among young adults. The longevity of school-based interventions needs stronger support (or qualification) and there are strong arguments for a more comprehensive approach that incorporates school based measures alongside population-level interventions. Could you develop the background to consider use of complementary strategies? I would also urge some caution about

young people's perceptions as the measures they support are not necessarily those they see as most effective. I also wondered if you could clarify your objectives, some of which seem to require a different design (e.g. (iii) seems to be better suited to a pre-post design).

Response: We appreciate the Reviewer's feedback. More details about smoking in China, especially in school contexts, have been incorporated in the 'Background' section. The discussion of complementary strategies has been added in the second last paragraph. We also incorporated some caution about young people's perceptions as our findings suggest that students obtained their knowledge about smoking in an agentic and active way. Objective (iii) was revised as suggested.

R2.4 Could you provide some justification for your use of focus groups with students – why not undertake in-depth interviews to elicit richer data, or use friendship dyads? Why select students who had previously participated in a longitudinal survey on smoking and some who had received a smokefree intervention? I can understand how the sample was convenient, but can you also explain how your approach provided a robust sample? How did you manage the social desirability error that you suggest might have impeded participants from commenting on their smoking status? How did you assess data saturation (four focus groups seems relatively few to achieve saturation)? Could you also explain the ethics review process you used? How did you manage translations? Can you be sure the people you interviewed had access to the information needed to answer your research questions (i.e., were teachers involved in evaluating smokefree schemes? Did they have input into the management of these?).

Response: We thank the Reviewer for this feedback. Justifications of using focus groups were added in the 'Data collection' section. As mentioned in the 'Sample' section, a brief evaluation of the intervention was conducted among participants who had intervention experience; thus, our sample included students of both conditions; all students completed the same surveys. More information about sample size, saturation, and social desirability error management is now included in the 'Data collection' section. Following BMJ Open's style, a description of the ethics review process has been incorporated at the end of the 'Data collection' section.

R2.5 I was a little surprised that you combined data from students and teachers, given they have quite different knowledge, power and perspectives, it might be more profitable to analyse the samples separately? Some themes draw only on one group (e.g. the ideological approach) and the theme is very descriptive with little probing of the tension between possible actions (e.g. expulsion) and actual actions taken.

There seems to be an emphasis on collective punishments (group demerit points) – could you comment on this approach compared to the individual interventions outlined earlier in the section?

Response: We thank Reviewer 2 for raising these points. As the comparison of students and teachers is the focus of our study, we retained the original setting out while detailing the differences between the groups. We also added more descriptions of the background and practical implications of collective punishments in both 'Background' and 'Discussion' sections.

R2.6 The theme on anti-smoking strategies not working seemed to cover several sub-themes, such as monitoring and enforcement; mis-understanding of smoking's risks and nature, and collective vs individual interventions. Can you tease out these quite different ideas a little more and comment on their implications?

Response: We thank the Reviewer for these suggestions in relation to the second theme. We now have used clearer sub-themes for better readability. Related implications were also included in the Discussion section.

R2.7 Evidence that tobacco was easily available suggests supply-side interventions and smokefree area measures could be appropriate. Could you discuss policy measures and how these could parallel or support some of the individual approaches you outlined? Often environmental change will bring about changes in social norms because some behaviours will no longer be possible. Could you develop discussion of policy interventions that could better support the work teachers are doing in schools?

Response: We appreciate these helpful suggestions from Reviewer 2 about policy intervention. We now have incorporated discussions about policy measures in the Discussion section (second last paragraph).

R2.8 Finally, as noted above, your discussion of limitations could be expanded to include topics other than sampling.

Response: We agree with Reviewer 2's suggestion in relation to expanding the limitations. This section now has been expanded to other possible methodological issues as well as future research directions.

I hope these comments are useful and I wish you well with future research.

Reviewer: 3

This manuscript reports on a qualitative study of students and teachers at two Chinese high schools about school-based smoking policies. I found this manuscript very interesting and enjoyable to read. I really appreciated the field notes that gave a sense of context to the findings for those not familiar with the environment in relation to tobacco control in China. I thought the sample size and the methods were appropriate, though further detail could be provided about various methodological aspects (which I've specifically discussed below). The authors illustrate very well how difficult it is to implement school-based anti-smoking strategies when the surrounding culture is not conducive to this. They also provide an insight into the mistaken beliefs of young people about tobacco, who downplay the health harms of smoking, e.g., the student who says that "Smoking moderately will not harm people."

As the authors point out, it is very important to conduct research on young people in China and their attitudes towards smoking, given the high and increasing smoking rates in this country, and the huge number of people who will suffer smoking-related disease in the future.

I think some aspects of the methods and results could be clearer and more focused, and have made suggestions below in this regard:

R3.1 On Page 4, in the background section, the authors state that prevalence of smoking in young people is increasing. It would be good to have some numbers here – what does the latest data say about the proportion of adolescents who smoke in China?

Response: We thank Reviewer 3 for the feedback about prevalence. We now have incorporated the smoking rates in the 'Background' section.

R3.2 In the abstract, it would be helpful to add that Kunming is in Yunnan Province, as you talk about high smoking prevalence in Yunnan Province in the introduction, but many people won't know that Kunming is the capital of Yunnan (and you don't make this clear until Page 7).

Response: We thank the Reviewer for pointing out this unclear statement. The description of the study's location has now been revised accordingly.

R3.3 Can you provide information about how the students were recruited? Did the teacher mention it to students and allow them to self-select, or did teachers/researcher select students based on certain characteristics?

Response: Consistent with Reviewer 3's suggestions, more details about recruitment information has been added in the 'Sample' section.

R3.4 Did you collect data on how many of the students and teachers were current smokers, or had tried cigarettes? This has obvious relevance for their attitudes, so it would be good to report, if you have this information.

Response: We thank the Reviewer for raising this interesting issue. Our study did not survey participants' smoking status. To clarify this issue, we have included the sentence "No demographic or smoking-related information was collected from interviewees".

R3.5 Some sentences are unnecessarily wordy and could be amended for better clarity. For example the first sentence in the methods reads "We used mixed methods with various samples to identify the inter-woven factors embedded in multi-layered phenomena and potentially avoid bias in homogenous sample." This could be stated much more clearly. As another example, on Page 6, Line 50: "Based on our research objectives, the research team developed guidelines to elicit factors that may have facilitated or hampered achievement of the desired outcomes." I'm not clear here what "desired outcomes" refers to, or what the "guidelines" are in relation to.

Response: We appreciate the Reviewer's feedback about better clarity for smoking of our sentences. Revisions to wordy sentences have been made.

R3.6 Were the interview questions the same or different for teachers and students? I assume they would be different questions, as they are quite different populations. The "general questions" listed on Page 6, Line 55 seem to be appropriate to ask teachers, but not students. For example, it's not really relevant to ask students "How will you improve the tobacco management at school", as this is not their responsibility. Providing more details about the interview schedule for both teachers and students would be helpful.

Response: We thank the Reviewer for pointing out this unclear statement in our original version. Those questions were asked for both students and teachers, because we were interested in the contrast of these two groups with distinct powers and status at school. We now have included a statement to clarify why the same questions were used for both groups.

R3.7 Relatedly, when reporting results, it needs to be drawn out more clearly whether the themes being discussed are based on data from the teachers or from the students.

Response: We thank Reviewer 3 for this feedback. Sentence subjects now have been revised to be specific as they represented distinct groups.

R3.8 On Page 7, Line 10, does "late teenagers" mean "older teenagers."?

Response: We have changed the wording as suggested.

R3.9 The concept of "ideological work" seems rather vague to me. From what you've described, it seems that the teachers have a range of strategies for dealing with smoking at school, including practical strategies such as patrolling dormitories and checking bags. There are also more educational strategies, which aim to teach children about the dangers of smoking, and also to educate

them about the rules regarding smoking at school. I'm just not sure that concept of "ideological work" adequately covers the data that the authors have presented.

Response: We appreciate this feedback from Reviewer 3. We have replaced the first theme with a clearer title. Also, to better clarify the meaning of ideological work and its background guidelines, a paragraph has now been added in the 'Background' section.

R3.10 Based on the findings presented there, the theme "Anti-smoking strategies at school are unlikely to work" might be better reframed in terms of pessimism about school based anti-smoking strategies.

Response: We thank Reviewer 3 for this suggestion and this theme name now has been updated.

R3.11 On Page 11, Line 44, the authors write that "The effectiveness of anti-smoking education was doubted by all participants. Instead of health promotion, safety was the ultimate reason behind tobacco control at school as smoking causes fires." It would be helpful here, and in the manuscript in general, to tease out a bit more which themes were raised by students and which were raised by teachers. These are very different population groups. So for example, instead of writing "All participants" you could write "Both students and teachers" to make it clear this was a theme for both groups.

Response: We agree with Reviewer 3's suggestion. Sentence subjects now have been revised to be specific as they represented distinct groups.

R3.12 How do cigarettes "alleviate embarrassment" (Page 13, Line 27). Can you elaborate on this?

Response: We thank the Reviewer for pointing out this unclear wording. We have now revised the wording. To better illustrate it, we have also included a quote.

R3.13 Page 15, Line 3, missing word. Should read "As school-based anti-smoking programmes have failed..."

Response: Thank you for raising this issue. We have inserted the missing word.

R3.14 On Page 15, Line 12, it would be more accurate to write that "...participants reported that both approaches failed to sufficiently curb students' smoking" as you are not actually measuring the schools' anti-smoking impact on smoking, just people's perception of it.

Response: We agree with Reviewer 3 and now have included 'participants reported' to improve the accuracy of the statement.

R3.15 Page 15, Line 20, should read "the ineffectiveness of schools' tobacco control programs." Or policies.

Response: Thank you. We now have rewritten the statement as suggested.

R3.16 On Page 15, Line 44, the authors write that "In this study, mechanisms that enabled tobacco control to be effective were only limited to the concern about safety." There isn't much data in the results about this. Could a quote or two be added to support this?

Response: As suggested by Reviewer 3, a quote now has been included.

R3.17 On Page 15-16 you talk about literature showing that school students smoke to cope with academic stress. This literature should also be mentioned in the introduction.

Response: Consistent with Reviewer 3's recommendation, we now have added some background information about this issue in the 'Background' section.

R3.18 On Page 16, Line 20, the sentence "Rather than establishing strict tobacco-free regimes at school which are currently unavailable, studies with longitudinal and other robust designs should pilot the effectiveness of these policies, because most associated research was conducted in Western countries and the effectiveness of tobacco management remains inconclusive." This sentence is convoluted and rather confusing. Can you reword for clarity?

Response: As suggested by the Reviewer, we have reworded this statement for clarity.

R3.19 On Page 16, Line 41 the authors write that "As previously found, the connection between students' smoking and coping with academic stress may be socially constructed." Can you elaborate more on what you mean by this?

Response: As suggested by Reviewer 3, the original wording has been replaced with a clearer statement.

R3.20 On Page 17, Line 26 the word "manage" should be "management."

Response: Thank you. We have changed the wording.

Attached is the revised paper as requested. Again, I would like to thank you for the useful feedback and for your attention to this manuscript.

VERSION 2 – REVIEW

REVIEWER	Charles Saunders Florida State University College of Medicine
REVIEW RETURNED	08-Dec-2017

GENERAL COMMENTS	In this revision the authors have made significant improvements to the manuscript. It is highly likely the finding that "Anti-smoking policy in Chinese schools is seemingly a well-structured system with education, monitoring, and enforcement processes. However, consistent with evaluations of the effectiveness of tobacco management in the West, the policy does not appear to be effective" is accurate due to the ease of teenagers purchasing cigarettes, having family members who smoke, seeing teachers smoke all coupled with a lax societal attitude toward smoking and the view that cigarettes are a socializing mechanism. These issues are common across the tobacco literature in virtually every country where cigarette use is not only tolerated but accepted. This does not mitigate the fact that the study is done with a non-probabilistic paid sample of 24 students and five teachers in two high schools in one region of China. Admittedly, qualitative analysis is difficult and the authors acknowledge that "Unlike positivistic
---

	research, our study did not aim to test predetermined hypotheses or create generalization...". Given these parameters, the study's results, as stated above, are consistent with the literature on adolescent tobacco usage.
--	--

REVIEWER	Kylie Morphet University of Queensland, Australia
REVIEW RETURNED	15-Dec-2017

GENERAL COMMENTS	The authors have done a good job of addressing reviewer comments. I have only a few very minor comments: In the first paragraph of the background, it should be made clear that the meta-analysis reports on adolescent smoking rates. Eg, "According to a meta-analysis, current smoking rates of male and female adolescents were estimated..." The authors state that "Student participants were recruited by the researcher at the end of the third wave survey" but they still do not state if this was directed recruitment (researcher selected students based on certain characteristics), or whether all students who were part of the survey were given the chance to participate in interviews. This could be clarified. The authors state that "No demographic or smoking-related information was collected from interviewees." The authors also state that "Both students and staff members frankly shared smoking-associated experiences of themselves or friends/family members." It would be more accurate to say that participants were not formally asked their smoking status, but that it sometimes/often arose during interviews.
--

VERSION 2 – AUTHOR RESPONSE

Reviewer: 1

Reviewer Name: Charles Saunders

Institution and Country: Florida State University, College of Medicine Please state any competing interests: None Declared

Please leave your comments for the authors below In this revision the authors have made significant improvements to the manuscript.

Comment: It is highly likely the finding that "Anti-smoking policy in Chinese schools is seemingly a well-structured system with education, monitoring, and enforcement processes. However, consistent with evaluations of the effectiveness of tobacco management in the West, the policy does not appear to be effective" is accurate due to the ease of teenagers purchasing cigarettes, having family members who smoke, seeing teachers smoke all coupled with a lax societal attitude toward smoking and the view that cigarettes are a socializing mechanism. These issues are common across the tobacco literature in virtually every country where cigarette use is not only tolerated but accepted.

This does not mitigate the fact that the study is done with a non-probabilistic paid sample of 24 students and five teachers in two high schools in one region of China. Admittedly, qualitative analysis is difficult and the authors acknowledge that "Unlike positivistic research, our study did not aim to test

predetermined hypotheses or create generalization...". Given these parameters, the study's results, as stated above, are consistent with the literature on adolescent tobacco usage.

Response: We thank Reviewer 1 for acknowledging the difficulties associated with the qualitative nature of our methodology. We agree that sampling is a limitation of our study and now have highlighted this limitation more explicitly in the 'Strengths and limitations of this study' section after the Abstract.

Reviewer: 3

Reviewer Name: Kylie Morphett

Institution and Country: University of Queensland, Australia Please state any competing interests: None

Please leave your comments for the authors below The authors have done a good job of addressing reviewer comments. I have only a few very minor comments:

Comment: In the first paragraph of the background, it should be made clear that the meta-analysis reports on adolescent smoking rates. Eg, "According to a meta-analysis, current smoking rates of male and female adolescents were estimated..."

Response: We thank Reviewer 3 for pointing out this unclear statement. This description has now been clarified.

Comment: The authors state that "Student participants were recruited by the researcher at the end of the third wave survey" but they still do not state if this was directed recruitment (researcher selected students based on certain characteristics), or whether all students who were part of the survey were given the chance to participate in interviews. This could be clarified.

Response: We thank Reviewer 3 for this feedback. Student participants were recruited from the students who previously finished the surveys and responded to an invitation to all students from the previous study to participate in interviews. We now have added this information for clarity.

Comment: The authors state that "No demographic or smoking-related information was collected from interviewees." The authors also state that "Both students and staff members frankly shared smoking-associated experiences of themselves or friends/family members." It would be more accurate to say that participants were not formally asked their smoking status, but that it sometimes/often arose during interviews.

Response: We thank Reviewer 3 for raising this issue. To keep the two statements consistent, we now have included an adverbial clause ("although participants were not formally asked about their smoking status") before "both students and staff members frankly shared smoking-associated experiences of themselves or friends/family members".

Attached is the revised paper as requested. Again, I would like to thank you for the useful feedback and for your attention to this manuscript.